# Density scaling of generalized Lennard-Jones fluids in different dimensions

Thibaud Maimbourg[1,2*], Jeppe C. Dyre[3], Lorenzo Costigliola[3*]

**1** The Abdus Salam International Centre for Theoretical Physics, Strada Costiera 11, 34151 Trieste, Italy
**2** LPTMS, CNRS, Université Paris-Sud, Université Paris-Saclay, 91405 Orsay, France
**3** Glass and Time, IMFUFA, Department of Science and Environment, Roskilde University, P.O. Box 260, DK-4000 Roskilde, Denmark.
* thibaud.maimbourg@lptms.u-psud.fr ; lorenzo.costigliola@gmail.com

## Abstract

Liquids displaying strong virial-potential energy correlations conform to an approximate density scaling of their structural and dynamical observables. This scaling property does not extend to the entire phase diagram, in general. The validity of the scaling can be quantified by a correlation coefficient. In this work a simple scheme to predict the correlation coefficient and the density-scaling exponent is presented. Although this scheme is exact only in the dilute gas regime or in high dimension $d$, a comparison with results from molecular dynamics simulations in $d = 1$ to 4 shows that it reproduces well the behavior of generalized Lennard-Jones systems in a large portion of the fluid phase.

# 1 Introduction

The past 20 years have developed an increasing interest in the so-called density-scaling approach. Starting from the experiments of Tölle, Dreyfus and Alba-Simionesco [1–3], it has been found that many liquids at different state points $(\rho, T)$ in their phase diagram exhibit a constant relaxation time along lines with fixed ratio $\rho^\gamma/T$. $T$ and $\rho$ are the temperature and density, respectively, and $\gamma$ is the so-called density-scaling exponent. This behavior has been often interpreted as the result of the repulsive part of the interaction potential giving the dominant contribution of the dynamics, allowing for mapping to an inverse-power-law (IPL) pair potential: $v_{\mathrm{IPL}}(r) = \varepsilon(\sigma/r)^\lambda$ [4–10]. Knowledge of the density-scaling exponent at a given state point allows one to transfer information about the dynamics of the system to other state points, thus drastically reducing the amount of experiments needed to measure properties of a system in its phase diagram. It allows as well predictions of the dynamics in regimes that are difficult to probe experimentally. For an IPL system the density-scaling exponent is related to the IPL exponent $\lambda$ through the relation $\gamma = \lambda/d$, where $d$ is the spatial dimension. The exponent is therefore constant throughout the entire phase diagram. As such, the IPL potential is paradigmatic as density scaling holds exactly. Then one may use this scaling approach for other systems. This has been successful in many situations [11–13] but is undermined by several weak points:

1. The very idea of mapping a real material's interaction to an IPL pair potential is not satisfying, as real materials display gas-liquid and gas-crystal coexistence not present in the phase diagram of the IPL system;

2. There are several evidences [13–16] that density scaling does not work for all systems. For real systems it cannot apply in the entire phase diagram and the density scaling exponent is state-point dependent [17–19];

3. It has been argued by Bøhling *et al.* [20] that the standard assumption that the repulsive and attractive parts of the potential play separate roles is difficult to justify.

A formulation of density scaling that eliminates the above-mentioned problems and contradictions is provided by the isomorph theory [11, 21]. In this framework, the density-scaling exponent $\gamma$ is generally state-point dependent. The invariance of static and dynamic properties along the lines of constant excess entropy (the so-called isomorphs), which defines the density-scaling exponent, is related to a scale invariance of the potential-energy hypersurface [22]. According to isomorph theory $\gamma$ can be obtained from equilibrium simulations at the state point in question [11]; for a real material it can also be determined by measuring several quantities at the same state point [17, 23].

The density-scaling exponent is not the only relevant state-point dependent quantity in the isomorph theory: it is paired with the virial potential-energy correlation coefficient $R \in [-1, 1]$, the value of which indicates whether or not density scaling is satisfied in the proximity of the state point in question. It can be shown that perfect invariance of structure and dynamics along isomorphs is ensured if the correlation coefficient is equal to unity, $R = 1$ [11]. This identity holds only for Euler homogeneous potentials [11], as IPL for instance, while for a real system $R = 1$ never applies. Therefore, a threshold value of 0.9 has been proposed: whenever $R > 0.9$, density scaling is expected to work well. Both quantities are defined

through equilibrium correlations of canonical-ensemble constant-volume fluctuations:

$$R = \frac{\langle \Delta W \Delta U \rangle}{\sqrt{\langle (\Delta W)^2 \rangle \langle (\Delta U)^2 \rangle}}, \qquad \gamma = \frac{\langle \Delta W \Delta U \rangle}{\langle (\Delta U)^2 \rangle} \tag{1}$$

in which $U$ denotes the potential energy, $W$ is the virial [1] (i.e. , the excess part of $PV$ with respect to the ideal gas, where $P$ is the pressure and $V$ the volume), and $\Delta$ is the instantaneous deviation of a given quantity from its thermodynamic equilibrium value.

The quantities $\gamma$ and $R$ are well defined in the entire phase diagram, not only where density scaling works. While $\gamma$ can be determined in experiments [6, 10, 17, 19, 23], this is not the case for $R$. It would therefore be useful to be able to relate the condition of $R$ crossing some threshold value to properties of the state-point dependence of $\gamma$, or even to be able to compute in a simple manner the state-point dependence of $R$. A first step in this direction was taken by Friisberg *et al.* [25], who showed that for generalized LJ potentials with exponents $(2n, n)$ – defined below, in Eq. (3) – the above-mentioned value $R = 0.9$ roughly coincides with the state points with highest density-scaling exponent $\gamma$. A convenient way of showing this is to plot $\gamma(\rho, T)$ versus $R(\rho, T)$. It was also shown that, in this system, $\gamma$ is a unique function of $R$: $\gamma(\rho, T) = F(R(\rho, T))$, to a good approximation. Although a physical interpretation for such a relation is not yet available, it allows one to better understand density scaling throughout the phase diagram. We emphasize that such a plot is related to the system *not* being a simple IPL system, as in the IPL case all state points map onto the single point $(R, \gamma) = (1, \lambda/d)$. While this diagram is *a priori* potential dependent (see [26, Fig. 3] for the $EXP$ potential), in the case of the LJ potential different thermodynamic phases are mapped into different regions of the diagram.

In the present work, generalized Lennard-Jones systems are studied theoretically and numerically, extending the results of Ref. [25]. The density-scaling quantities mentioned above are derived from excess thermodynamic observables, which are naturally expressed in the language of the virial expansion [24]. We devise a simple low-density approximation for the density-scaling exponent and the virial-potential energy correlation coefficient, and compare it to computer simulations, with very good agreement. Then, as this approximation becomes also exact in the limit of infinite dimension, we connect these results to the recent finding that isomorph invariance is exactly achieved for a large class of potentials as $d \to \infty$, beyond the Euler-homogeneous ones like IPLs, in the liquid and glass phases [27]. The generalized Lennard-Jones potentials, albeit very important in practice, do not belong to this class; but computer simulations in Ref. [28] provided evidence that density-scaling becomes more robust when increasing the number of dimensions, including for state points close to the liquid-gas coexistence. We thus clarify and extend the results of Refs. [27, 28]. We observe a convergence to the large-$d$ limit, starting already from dimension $d = 2$.

The structure of the paper is the following. In the next section we introduce the system studied and summarize past observations relevant to the present article. In the third section we present the analytic approximation which is compared in the fourth section to molecular dynamics simulations in two, three, and four dimensions, over a wide range of temperatures and densities. A concluding discussion ends the paper.

---

[1] For a system of $N$ particles at positions $\{\mathbf{r}_1, \ldots, \mathbf{r}_N\}$ in equilibrium, one has $PV = k_B N T + \langle W \rangle$ [24], where

$$W = -\frac{1}{d} \sum_{i=1}^{N} \mathbf{r}_i \cdot \nabla_{\mathbf{r}_i} U \tag{2}$$

## 2 Previous results on generalized Lennard-Jones potentials

Following Friisberg *et al.* [25], the class of generalized Lennard-Jones pair potentials is studied in this work. It is defined as follows:

$$v_{m',n'}(r) = \frac{\epsilon}{m'-n'} \left[ n' \left( \frac{\sigma}{r} \right)^{m'} - m' \left( \frac{\sigma}{r} \right)^{n'} \right] \tag{3}$$

and denoted by $LJ(m', n')$. These are also known as Mie potentials [29], providing a generalization of the standard 12-6 LJ potential that is widely used in the liquid-state literature. The $LJ(m', n')$ potential is displayed in Fig. 1 for several choices of the integers $(m', n')$.

The main reasons for the normalization factors of the two IPLs are:

1. The minimum of the $LJ(m', n')$ potential is located at $r = \sigma$ where the potential's value is $-\epsilon$, independently of $(m', n')$. This facilitates comparison between different potentials in the class and gives a simple low-density picture of particles with a constant effective diameter $\sigma$ with repulsive and attractive forces that becomes steeper and shorter ranged as exponents are increased, see Fig. 1.

2. One recovers the standard three-dimensional 12-6 LJ with a simple rescaling of $(\varepsilon, \sigma)$ in the potential $LJ(12, 6)$.

3. As discussed below, for large $d$ the exponents $(m', n')$ must be scaled linearly with $d$. The normalizations are such that this scaling impacts the IPL exponents while the factors independent of $r/\sigma$ are invariant. This allows for a well-defined comparison between different dimensions.

In this work the exponents $m'$ and $n'$ are changed independently, generalizing the treatment of Ref. [25]. We shall consider only integer , but the conclusions do not depend on this limitation.

For convenience we define the ratio

$$X = \frac{m'}{n'} \qquad (X > 1). \tag{4}$$

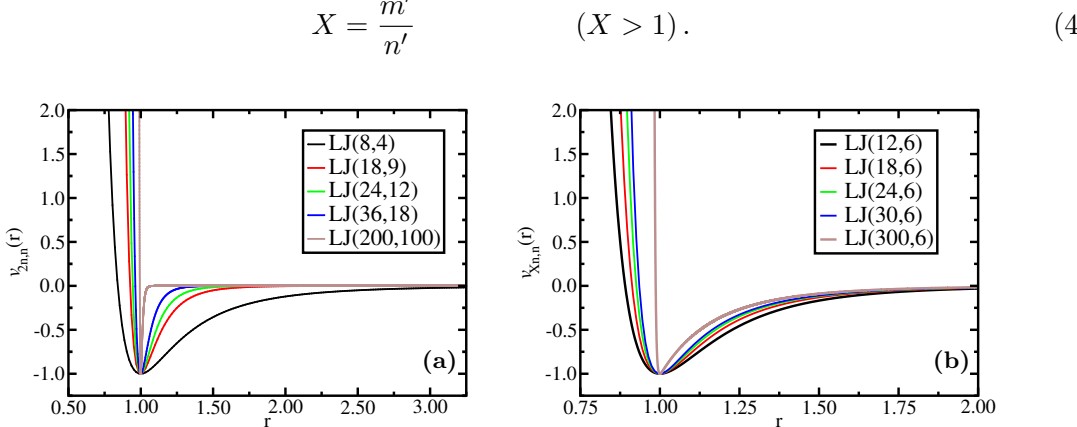

Figure 1: Different generalized $LJ(m', n')$ pair potentials. (a) $m' = 2n'$ for several choices of the value of $n' = 4, 9, 12, 18, 100$. Upon increasing $n'$ the potential becomes sharper and sharper, approaching a sort of sticky-sphere limit (similar in spirit to the discontinuous version of Baxter [30]). (b) $m' = Xn'$ for $X = 2, 3, 4, 5, 50$ and fixed $n' = 6$. For large $X$ the attractive tails of the potential tend to a limiting shape, while the repulsive part approaches a hard wall at $r/\sigma = 1$.

The values $X \leqslant 1$ are excluded because they do not define a physical liquid pair potential. Note that at low density for fixed temperature (or high temperature for fixed density) the repulsive IPL term dominates, implying $R \to 1$, $\gamma \to m'/d$. Although in general the correlation coefficient is below unity, it was shown in Refs. [11, 14, 31] that the standard 12-6 LJ potential is strongly correlating (i.e. , obey $R > 0.9$) in the region above the freezing line in the $(\rho, T)$ phase diagram. Furthermore, in Ref. [25] the LJ$(2n', n')$ potential was studied at many state points in dimensions two to four, displaying either gas, liquid/fluid, crystal phases or coexistence between these. Away from strongly-correlating regions, it was found that the simple relation

$$\gamma \simeq 3n'R/d \qquad (5)$$

holds to a good approximation when varying dimensions $d$ or exponent $n'$. As pointed out in Sec. 1, such a relation is most welcome in order to assess quickly isomorph invariance, controlled by the $R$ value, from an experimentally measured $\gamma$. An analytically manageable formula for $R$ and $\gamma$ would be useful for better understanding why and how these two quantities correlate.

The simple dimensional dependence, Eq. (5), similar to the prediction of an IPL system with exponent $3n'$ [31], is intriguing. It was shown in Ref. [27] that in the dense liquid regime, a sizeable class of potentials are effectively IPLs in large dimension and as such display isomorph invariance. They are therefore of little use for understanding the variations of $(R, \gamma)$ observed in real systems. This is, however, not the case for the LJ$(m', n')$ potentials [27], which retain their 'non-IPL' behavior even in high dimensions. For this limit to be well-defined, one needs to scale the exponents with the dimension and thus consider LJ$(md, nd)$ pair potentials[2] [27, 35–37]. In this case, one expects the large-dimensional limit to exhibit scaling properties closer to the finite-dimensional systems than the potentials studied in Ref. [27], while still being considerably simpler.

Molecular dynamics simulations in Ref. [28] showed that the standard 12-6 LJ fluid above (in density or temperature) the liquid-vapor critical point manifests increasingly better isomorph invariance as $d$ goes from 2 to 4, i.e. , has here an increasing $R$. This observation tends to broaden the conclusions of Ref. [27], which hold in the liquid and glassy regimes, to the fluid region close to the liquid-vapor critical point, which cannot be described by the arguments developed in Ref. [27]. Indeed they rely on a truncation of the virial expansion to the second virial coefficient, valid for dense liquid regimes in high $d$ [38–40], whereas the third virial coefficient cannot be neglected close to the liquid-vapor critical point [24]. It turns out that, in the infinite-dimensional limit, the liquid-vapor critical point regime and the dense (possibly supercooled) liquid regimes occur at density ranges exponentially separated in $d$ [41], displaying different behaviors. We point out that in these simulations (Ref. [28]), the exponents were not scaled with $d$, which is mandatory to compare with high-$d$ theory. We take this into account in the present work, which investigates the effect of dimensionality in light of a systematic comparison to the mean-field prediction for LJ$(md, nd)$ interactions.

---

[2]The exponents must be scaled with $d$ for the thermodynamic limit to be well defined [32, 33]. At large $r$ the potential must vanish faster than $r^{-d}$, so $n > 1$ for the interaction to be short ranged. This scaling of the exponents with dimensions also preserves the slope of the freezing line, at least in the high-density limit [11, 28, 34].

# 3 Analytic expressions derived from the virial expansion

In order to get an analytic handle[3] on the quantities $\gamma$ and $R$, we study here their lowest order in the virial expansion of liquid-state theory [24]. The reason is threefold: First, the quantities of interest are excess thermodynamic observables, for which this framework has been designed. One expects that this expansion is well suited to the study of gases and dilute liquids as the density is small, while it could perform poorly for dense liquids in low dimensions where other perturbation strategies or re-summations work better [24]. Second, it is also the right choice for the mean-field limit of large dimension [37–40, 44], allowing us to observe the impact of $d$. Third, the observed density dependence of the $(R, \gamma)$ diagram is found to be very mild and one may thus hope that the virial approximation provides a good starting point for future more accurate approaches.

## 3.1 The lowest order in the virial expansion in any dimension

In this section we give the first-order virial expansion of certain thermodynamic fluctuation averages. The number density $\rho = N/V$ is regarded as a small parameter. The lowest order of any observable is then its ideal-gas value; we are interested in the first non-trivial small-density correction. The equilibrium fluctuations computed below are the ones entering in the definition of the virial-potential energy correlation coefficient $R$ and the density-scaling exponent $\gamma$, i.e. , the fluctuations $\langle \Delta U \Delta W \rangle$, $\langle (\Delta U)^2 \rangle$ and $\langle (\Delta W)^2 \rangle$ (Eq. (1)). We shall here just give the main steps as a detailed derivation is found in [27, Appendix A]. Using the definition of canonical equilibrium averages, the first two fluctuations are rewritten as follows,

$$\langle \Delta W \Delta U \rangle = \langle WU \rangle - \langle W \rangle \langle U \rangle = -\frac{\partial \langle W \rangle}{\partial \beta}$$

$$\langle (\Delta U)^2 \rangle = -\frac{\partial \langle U \rangle}{\partial \beta} = -\frac{\partial^2 (\beta F)}{\partial \beta^2} \tag{6}$$

where $\beta = 1/T$ is the inverse temperature (we set the Boltzmann constant to unity, thus measuring temperature in energy units) and $F$ the Helmholtz free energy.

The average virial is related to the pressure by the virial equation of state [24]. The expansion of the latter and of the free energy follows from a standard computation [24, 39, 40], with $v(r)$ being the radial pair potential:

$$\frac{\beta F}{N} = \ln \rho - 1 - \frac{\rho}{2} \int d\mathbf{r} \left( e^{-\beta v(r)} - 1 \right) + O(\rho^2)$$

$$\frac{\beta P}{\rho} = 1 + \beta \frac{\langle W \rangle}{N} = 1 - \frac{\rho}{2} \int d\mathbf{r} \left( e^{-\beta v(r)} - 1 \right) + O(\rho^2) \tag{7}$$

Combining Eqs. (6) and (7) we get expressions for $\langle \Delta W \Delta U \rangle$ and $\langle (\Delta U)^2 \rangle$. The derivation of $\langle (\Delta W)^2 \rangle$ from the lowest-order virial expansion of the two-point distribution function is

---

[3]We note that an analytical treatment via a transfer matrix method is possible in $d = 1$ if one truncates the interaction range suitably (see e.g. Refs. [42, 43]).

slightly more involved [27, Appendix A]. The final result for all three fluctuations is

$$
\begin{aligned}
\langle (\Delta U)^2 \rangle &= \frac{N\rho}{2} \int d\mathbf{r} \, (v(r))^2 e^{-\beta v(r)} + O(\rho^2) \\
\langle (\Delta W)^2 \rangle &= \frac{N\rho}{2d^2} \int d\mathbf{r} \, (rv'(r))^2 e^{-\beta v(r)} + O(\rho^2) \\
\langle \Delta W \Delta U \rangle &= \frac{\rho N}{2\beta^2} \int d\mathbf{r} \left( e^{-\beta v(r)} - 1 \right) + \frac{\rho N}{2\beta} \int d\mathbf{r} \, v(r) e^{-\beta v(r)} + O(\rho^2) \\
&= \frac{\rho N}{2d\beta} \int d\mathbf{r} \left[ rv'(r) + dv(r) \right] e^{-\beta v(r)} + O(\rho^2)
\end{aligned}
\tag{8}
$$

where we performed an integration by part in the last line of Eq. (8). We now get the following from the definition Eq. (1):

$$
\begin{aligned}
R &= -\frac{1}{\beta} \frac{\int d\mathbf{r} \, [rv'(r) + dv(r)] e^{-\beta v(r)}}{\sqrt{\int d\mathbf{r} \, (rv'(r))^2 e^{-\beta v(r)}} \sqrt{\int d\mathbf{r} \, v(r)^2 e^{-\beta v(r)}}} + O(\rho) \\
\gamma &= -\frac{1}{\beta d} \frac{\int d\mathbf{r} \, [rv'(r) + dv(r)] e^{-\beta v(r)}}{\int d\mathbf{r} \, v(r)^2 e^{-\beta v(r)}} + O(\rho)
\end{aligned}
\tag{9}
$$

These first-order virial expressions apply for any isotropic pair-potential liquid.

A check of the validity of Eq. (9) can be obtained by considering the IPL potential $v_{\mathrm{IPL}}(r) = \varepsilon(\sigma/r)^\lambda$ for which we already know the result. Since $rv'_{\mathrm{IPL}}(r) = -\lambda v_{\mathrm{IPL}}(r)$, IPL systems have perfect virial-potential energy correlations ($R = 1$). Using spherical coordinates and standard properties of the Euler Gamma function one arrives at

$$
\begin{aligned}
R_{\mathrm{IPL}} &= \frac{\lambda - d}{\beta\lambda} \frac{\int_0^\infty dr \, r^{d-1} v_{\mathrm{IPL}}(r) e^{-\beta v_{\mathrm{IPL}}(r)}}{\int_0^\infty dr \, r^{d-1} v_{\mathrm{IPL}}(r)^2 e^{-\beta v_{\mathrm{IPL}}(r)}} \\
&\underset{x := \beta\varepsilon(\sigma/r)^\lambda}{=} \left(1 - \frac{d}{\lambda}\right) \frac{\int_0^\infty dx \, x^{-d/\lambda} e^{-x}}{\int_0^\infty dx \, x^{1-d/\lambda} e^{-x}} = \left(1 - \frac{d}{\lambda}\right) \frac{\Gamma\left(1 - \frac{d}{\lambda}\right)}{\Gamma\left(2 - \frac{d}{\lambda}\right)} = 1 \\
\gamma_{\mathrm{IPL}} &= \frac{\lambda - d}{\beta d} \frac{\int_0^\infty dr \, r^{d-1} v_{\mathrm{IPL}}(r) e^{-\beta v_{\mathrm{IPL}}(r)}}{\int_0^\infty dr \, r^{d-1} v_{\mathrm{IPL}}(r)^2 e^{-\beta v_{\mathrm{IPL}}(r)}} = \frac{\lambda}{d} R_{\mathrm{IPL}} = \frac{\lambda}{d}
\end{aligned}
\tag{10}
$$

which is the general result for IPL systems in $d$ dimensions [11], as mentioned in the introduction. It must be recovered from Eq. (10) because this general results holds in particular at low densities where the virial approximation becomes exact.

In the case of the LJ($md, nd$) potential in $d$ dimensions, i.e. , Eq. (3) with $m' = md$ and $n' = nd$, the quantities $R$ and $\gamma$ can be simplified further from Eq. (9). The result is obtained using similar considerations to the IPL case, e.g.,

$$
\begin{aligned}
\int d\mathbf{r} \, v_{md,nd}(r)^2 e^{-\beta v_{md,nd}(r)} &= \left(\frac{\varepsilon}{m-n}\right)^2 \Omega_d \int_0^\infty dr \, r^{d-1} \left[ n\left(\frac{\sigma}{r}\right)^{md} - m\left(\frac{\sigma}{r}\right)^{nd} \right]^2 \\
&\qquad \times e^{-\frac{\beta\varepsilon}{m-n}\left[n\left(\frac{\sigma}{r}\right)^{md} - m\left(\frac{\sigma}{r}\right)^{nd}\right]} \\
&\underset{y := (\sigma/r)^{nd}}{=} n\left(\frac{\varepsilon}{m-n}\right)^2 \mathcal{V}_d(\sigma) \int_0^\infty dy \, y^{1-\frac{1}{n}} (y^{X-1} - X)^2 e^{-\frac{\beta\varepsilon}{X-1}(y^X - Xy)}
\end{aligned}
\tag{11}
$$

in which $X = m/n$, $\mathcal{V}_d(r)$ is the volume of an hypersphere of radius $r$ in $d$-dimensional space, and $\Omega_d$ is the solid angle given by

$$\mathcal{V}_d(r) = \frac{\Omega_d}{d} r^d \;, \qquad \Omega_d = \frac{2\pi^{d/2}}{\Gamma(d/2)} \tag{12}$$

Note that the condition $n > 1$ discussed in footnote 2 implies that the $y^{1-1/n}$ divergence at small $y$ (long distances) is integrable (as well as $y^{-1/n}$), whereas our restriction $X > 1$ implies integrability at large $y$ (short distances). Similar considerations for the other integrals involved lead to the following expressions for $R$ and $\gamma$

$$R = \frac{X-1}{\beta\varepsilon X n} \frac{\int_0^\infty \mathrm{d}y\, y^{-\frac{1}{n}} \left[(Xn-1)y^{X-1} - X(n-1)\right] e^{-\frac{\beta\varepsilon}{X-1}(y^X - Xy)}}{\sqrt{\int_0^\infty \mathrm{d}y\, y^{1-\frac{1}{n}}(y^{X-1}-X)^2 e^{-\frac{\beta\varepsilon}{X-1}(y^X-Xy)} \int_0^\infty \mathrm{d}z\, z^{1-\frac{1}{n}}(z^{X-1}-1)^2 e^{-\frac{\beta\varepsilon}{X-1}(z^X-Xz)}}}$$

$$\gamma = \frac{X-1}{\beta\varepsilon} \frac{\int_0^\infty \mathrm{d}y\, y^{-\frac{1}{n}} \left[(Xn-1)y^{X-1} - X(n-1)\right] e^{-\frac{\beta\varepsilon}{X-1}(y^X-Xy)}}{\int_0^\infty \mathrm{d}y\, y^{1-\frac{1}{n}}(y^{X-1}-X)^2 e^{-\frac{\beta\varepsilon}{X-1}(y^X-Xy)}} \tag{13}$$

in which $y = z = (\sigma/r)^{nd}$. These equations establish the lowest order in the small-density expansion. We note that:

- $R$ and $\gamma$ are to lowest order *independent of the density* (keeping in mind that its validity is guaranteed only for low enough density). Only the temperature and the details of the potential enter as parameters. This is a direct consequence of the fact that they are expressed as ratios of extensive quantities. Indeed, any extensive observable $\mathcal{O}$ scales in the thermodynamic limit as $\langle \mathcal{O} \rangle \propto N \propto \rho$ and thus $\langle \mathcal{O} \rangle = O(\rho)$ in the virial expansion.

- $R$ and $\gamma$ are also *independent of the space dimensionality*. This is due to the fact that the potential is built with IPLs whose exponents are proportional to $d$. The numerical computation of these quantities in any dimension is therefore straightforward.

- The explicit $n$ dependence of $R$ and $\gamma/n$ is rather mild, unlike the $X$ dependence. This observation relies on the two limits: *(i)* for low enough temperature, the main contributions to the integrals in Eq. (13) come from the vicinity of the saddle point $y = 1$ *(ii)* for large enough temperature, one approaches the repulsive-IPL result $(R, \gamma/n) = (1, X)$. Fig. 2 gives an additional numerical verification.

These facts imply that a comparison to simulation data in any dimension is easily achieved. This is yet another reason to scale LJ($md$,$nd$) with dimension $d$ in this way, in addition to ensuring a well-defined thermodynamic limit. Besides, we expect any thermodynamic observable constructed as a ratio between extensive quantities involving only the pair potential $v_{md,nd}(r)$ to exhibit the properties mentioned in the above first two points.

As the dependence upon $n$ is quite mild, one can simplify the analytical expressions by

considering the large $n$ limit at fixed $X$. The three different integrals become

$$\int_0^\infty \mathrm{d}y\, y^{-\frac{1}{n}} \left[(m-1)y^{X-1} - X(n-1)\right] e^{-\frac{\beta\varepsilon}{X-1}(y^X - Xy)} \sim m \int_0^\infty \mathrm{d}y\, \left[y^{X-1} - 1\right] e^{-\frac{\beta\varepsilon}{X-1}(y^X - Xy)}$$

$$= -\frac{m}{\beta\varepsilon}\frac{X-1}{X}\int_0^\infty \mathrm{d}y\, \frac{\mathrm{d}}{\mathrm{d}y} e^{-\frac{\beta\varepsilon}{X-1}(y^X - Xy)} = \frac{m}{\beta\varepsilon}\frac{X-1}{X}$$

$$I_U(X,\beta\varepsilon) \equiv \int_0^\infty \mathrm{d}y\, y(y^{X-1} - X)^2 e^{-\frac{\beta\varepsilon}{X-1}(y^X - Xy)}$$

$$I_W(X,\beta\varepsilon) \equiv \int_0^\infty \mathrm{d}y\, y(y^{X-1} - 1)^2 e^{-\frac{\beta\varepsilon}{X-1}(y^X - Xy)}$$

$$(14)$$

The corresponding values of $R$ and $\gamma$ are

$$R \underset{n\to\infty}{\sim} \frac{1}{(\beta\varepsilon)^2}\frac{(X-1)^2}{X}\frac{1}{\sqrt{I_U(X,\beta\varepsilon)I_W(X,\beta\varepsilon)}}$$

$$\frac{\gamma}{m+n} \underset{n\to\infty}{\sim} \frac{1}{(\beta\varepsilon)^2}\frac{X-1}{X+1}\frac{1}{I_U(X,\beta\varepsilon)} \tag{15}$$

Both quantities are of order 1 with respect to $n$, which is why we divided $\gamma$ by a quantity proportional to $n$; we chose $m+n$ for better comparison with Ref. [25] as the linear relation found there reads $\gamma = (m+n)R$.

Finally, in the case of the LJ$(2nd, nd)$ potential ($X = 2$), the integrals become Gaussian and one arrives at the simple expressions

$$R(X=2) \underset{n\to\infty}{\sim} \frac{2}{\sqrt{G_U(\beta\varepsilon)G_W(\beta\varepsilon)}} \ , \qquad \frac{\gamma(X=2)}{n} \underset{n\to\infty}{\sim} \frac{4}{G_U(\beta\varepsilon)}$$

$$G_U(x) \equiv 2(1+x) + e^x(2x-1)\sqrt{\pi x}\left[1 + \mathrm{erf}(\sqrt{x})\right]$$

$$G_W(x) \equiv 2 + e^x\sqrt{\pi x}\left[1 + \mathrm{erf}(\sqrt{x})\right] \tag{16}$$

$R$ is a strictly increasing function of temperature, implying that $\gamma(T)$ can be parameterized instead by $R$ and the function $\gamma(R)$ is well defined. Its curve for $X = 2$ and any $n$ from Eq. (13) can as well be drawn by a parametric temperature plot. It has a non-trivial shape that is well described by the above $n \to \infty$ expression, as documented by Fig. 2. In Sec. 4 we confirm this by comparing the analytical results to simulation data.

## 3.2 The high-dimensional limit

The considerations of the last section are independent of the dimension $d$, but hold only in the dilute regime of the liquid phase. The virial truncation is, however, exact in the whole (possibly supercooled) liquid phase for mean-field situations, such as when the spatial dimension goes to infinity [37–40, 44], or in the infinite-range Mari-Kurchan model considering random shifts of the particles in any dimension [45]. We thus expect smaller deviations from the exact $d \to \infty$ reference situation as the dimension increases, but one does not know *a priori* how large the dimension must be to ensure satisfactory convergence to the mean-field prediction. This is the focus of a following section, Sec. 4.2.

We now discuss what can be expected from Eq. (13), focusing on the limit $d \to \infty$. Let us first consider the liquid phase. For large $d$ the physics of the LJ$(md, nd)$ potential (3) is

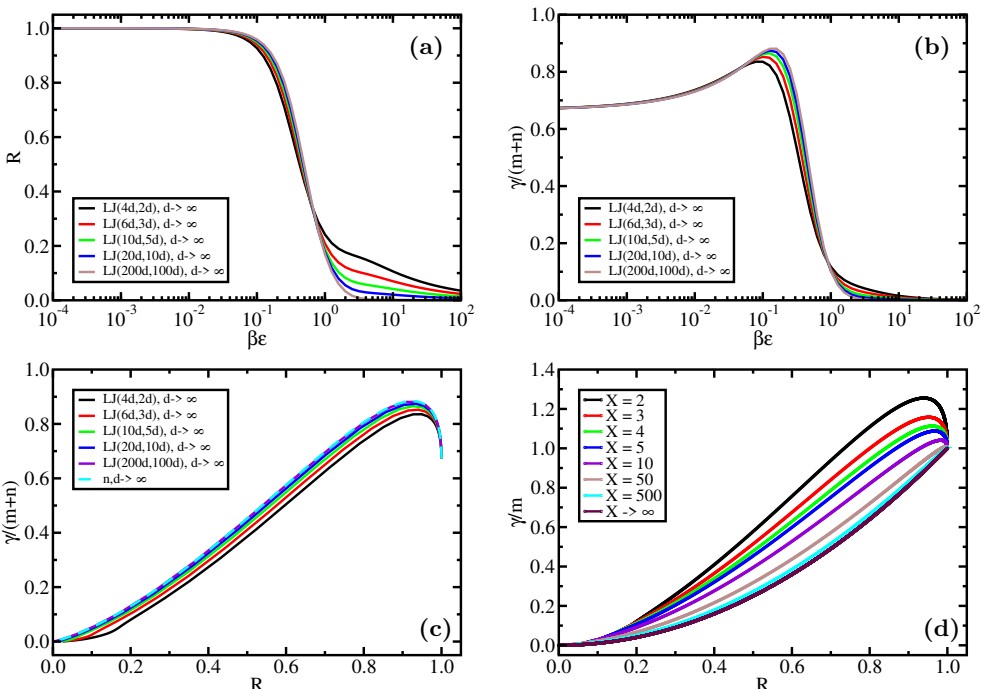

Figure 2: Results from the analytic expressions Eq. (13) derived in the first-order virial approximation (hereafter labelled $d \to \infty$, see 3.2) in any dimension $d$. (a)(b)(c) examine LJ($2nd, nd$) potentials with varying $n = 2 \to \infty$. The temperature dependence of $R$ and $\gamma$ is plotted in (a) and (b); the parametric plot $\frac{\gamma}{m+n}(R)$ is displayed in (c). The variation of $n$ weakly impacts these curves, well approximated by the simple $n \to \infty$ functions Eq. (16) (dashed light cyan curve in (c), coinciding with $n = 100$). Conversely, the $X$ dependence is stronger as shown in panel (d) for LJ($Xnd, nd$) potentials with fixed $n = 2$ and $X = 2 \to \infty$. This is commented further and confronted to simulations in Sec. 4.1 and Fig. 4. The limiting case of $X \to \infty$ leads to a parabolic dependence for $\gamma$, as shown in App. A.

as follows [27]. Both thermodynamics and dynamics are dominated by a fluctuating region of order $1/d$ around a length scale $r^*$ defined by requiring $\beta v_{md,nd}(r^*)$ to be of order unity with respect to $d$. The length scale $r^*$ plays the role of an effective particle diameter. At temperatures exponentially large in $d$, $r^* < \sigma$ and the attractive IPL term is exponentially suppressed. The system is then dominated by the repulsive IPL term and has exact isomorphs: $R = 1$, $\gamma = m$ (compare Eq. (10)). Conversely, for exponentially small temperatures, the system will be dominated by the attractive IPL term, which has a non-stable thermodynamic limit. The non-trivial regime, closer to the finite-dimensional system, is that of $O(1)$ temperature in which $r^* = \sigma$. In this case both terms in the potential compete and the potential has the following behavior

- for $r < \sigma$, the interaction is effectively hard core: $\beta v_{md,nd}(r) \to \infty$

- in the $1/d$ region around $\sigma$, i.e. , when $r = \sigma(1 + \tilde{r}/d)$ with $\tilde{r}$ a reduced distance of order unity, the potential is a sum of two competing repulsive and attractive exponentials

$$\beta v_{md,nd}(r) \sim \frac{\beta \epsilon}{m - n} \left( n e^{-m\tilde{r}} - m e^{-n\tilde{r}} \right) \tag{17}$$

- for $r > \sigma$, there is effectively no interaction and $\beta v_{md,nd}(r)$ is exponentially vanishing.

As in finite dimensions, both the attractive and the repulsive terms contribute in this regime. Consequently, no exact isomorphs are found, i.e., no rescaling of the density and the temperature can make the free energy and dynamics invariant [27]. One has $R < 1$ except in the infinite-temperature limit where $R \to 1$, which is the case in which only the repulsive IPL term dominates. The precise values and evolution of $R$ are provided in the next section.

From Eq. (16) it is clear that at low temperatures $(R, \gamma) \to (0, 0)$, and $R$ increases monotonically to 1 at high temperatures. Yet, as shown in the next section, there exist negative values of $R$ and $\gamma$, or large values of $\gamma$ at small $R$, as outcome of simulations in $d \leqslant 4$ due to state points in a liquid-vapor or solid-vapor phase coexistence. Such values are absent from Eq. (9). Indeed, even in large dimensions the validity of the latter equations is restricted to the uniform liquid phase, and Eq. (9) do not necessarily extend to the crystal or liquid-vapor coexistence regimes[4]. The reason is that in deriving the above equations we implicitly assumed that the thermodynamic density profile is uniform. On the one hand, while crystalline phases and the liquid-crystal transition have been studied for $d = 4, 5, 6$ in Refs. [46, 47], precise descriptions of large-$d$ crystalline phases are not yet possible as the equilibrium crystalline configurations for large $d$ are not known. This is due to the daunting geometry of spherical packings in high $d$ [48] and because numerically nucleating the crystal phase through compression of the liquid has proven extremely difficult when the dimension is larger than three [47, 49]. On the other hand, following a Landau approach for large $d$, the liquid-vapor coexistence regime may be studied by including one additional order in density in the virial expansion [24]. But then the density window considered is exponentially separated from the one of the liquid/fluid phase [39, 40], which makes comparison to the finite-d system harder. Indeed, the regime close to the critical point has been studied by Mon and Percus [41] for a square-well potential, where one can analytically extract a critical-point density whose exponential dependence is given through the effective packing fraction[5] $\varphi_c \approx (4/\sqrt{3})^{-d} \ll 2^{-d}$, much lower than the (dense) liquid phase scaling $\varphi = O(d/2^d)$ [37, 44]. The critical-point temperature is moreover not of order unity, as $T_c = O(1/d)$. We expect the same dimensional scaling of the critical density to hold for a LJ$(md, nd)$ potential.

# 4    Molecular Dynamics simulations in dimension one to four

In this section the expressions derived from the virial expansion are compared with Molecular Dynamics simulation data for different LJ potentials in $2d$, $3d$, and $4d$. These results include the standard 12-6 LJ potential in three dimension. The variation with the main parameters – the density, the potential exponents, and the dimension – will be analyzed. We find qualitative (if not quantitative) agreement with the analytic expressions of $R$ and $\gamma$, (Eq. (13)), in a large density regime.

Two different Molecular Dynamics codes were put to work. The $3d$ simulations employed the GPU-based Roskilde University Molecular Dynamics (RUMD) code [50], while the $1d$, $2d$ and $4d$ simulations used an *ad hoc* CPU-based code (more details can be found in Ref. [51]).

---

[4]We shall not consider glassy configurations in this article, although computations in metastable glasses are in fact possible, at least in the large-$d$ limit [37].

[5]The effective packing fraction means here the ratio of the volume occupied by the particles, defined effectively by spheres of radius $r^* = \sigma$, over the volume of the system.

All simulations were performed in the $NVT$ ensemble in which temperature was controlled using the standard Nosé-Hoover thermostat [52]. The time step $\Delta t$ is kept constant in reduced units ($\Delta \tilde{t}$) when exploring the phase diagram, i.e. , it gets rescaled at each state point via the only potential-independent time unit of the system: $\Delta t = \rho^{-1/d}(M/T)^{1/2}\Delta \tilde{t}$, in which $M$ is the particle mass (set to unity). In particular, this scaling ensures that the ballistic regime of the mean-squared displacement collapses onto a single master curve for all state points when distances are measured in units of $\rho^{-1/d}$. The reduced time step values, $\Delta \tilde{t}$, are given in Tab. 1 together with the system size $N$ and the number of simulation steps for each dimension. The potentials studied in this work were cut-off at distance $r_{cut} = 2.5\sigma$. The choice of the parameters in Tab. 1 is mainly dictated by running time limitations. Our *ad hoc* code is less optimized than RUMD and simulations cannot be run as extensively. We checked that the duration of the most time-consuming $4d$ simulations did not hinder accurate calculation of the thermodynamic quantities. In $4d$ the system size is chosen as the smallest that permits a reliable study of the highest-density state points ($\rho = 1.5$) in a reasonable time. This is detailed in App. B. In $3d$ simulations, two different system sizes are listed to ensure the absence of system-size dependence. Similarly, in $1d$ the results are reported for $N = 40000$ particles, but other simulations were run for smaller system size ($N = 500$).

The units of length and energy, respectively $\sigma$ and $\epsilon$, are set to 1. At each state point ($\rho, T$) we compute the correlation coefficient $R$ and the scaling exponent $\gamma$ from their definitions Eq. (1). Density lies within the interval $\rho \in [0, 1.5]$ and temperature $T \in [0.5, 5]$, which is enough to retrieve all relevant values of $R$ and $\gamma$. For a few isochores in $2d$ some simulations at higher temperature were carried out as detailed in Fig. 3's caption. In $3d$, data for the standard LJ system are also reported. A complete list of the state points studied in this work can be found at the data repository of the Glass and Time group (see Acknowledgements).

| $d$ | $N$ | $\Delta \tilde{t} \ (10^{-3})$ | $N_{\text{steps}} \ (10^6)$ |
|---|---|---|---|
| 1 | 40000 | $0.25 - 0.5$ | 20 |
| 2 | 1600 | 1 | 10 |
| 3 | $1000 - 4096$ | 1 | 500 |
| 4 | 2401 | 4 | 5 |

Table 1: Simulation parameters: The system size $N$, the reduced time step $\Delta \tilde{t}$, and the number of time steps simulated at each state point for all four dimensions considered numerically in this work. For the LJ(300, 6) potential, a smaller reduced time step $\Delta \tilde{t} = 10^{-4}$ was used because of the potential steepness at small distances.

Some of the state points exhibit phase coexistence; these are associated with low values of $R$ [12]. The density-scaling exponent $\gamma$ is not uniquely defined when different phases coexist. We find large $\gamma$ and small $R$ in the crystal-vapor coexistence, whereas in the liquid-vapor coexistence both $R$ and $\gamma$ are small (close to zero or negative). As mentioned in Sec. 3.2, state points in the coexistence region cannot be interpreted with the analytical equations obtained in this work because these equations were derived based on the implicit assumption of a single, isotropic phase. We therefore left out these state points in the following analysis except in Fig. 3 (c), where they are retained as an example.

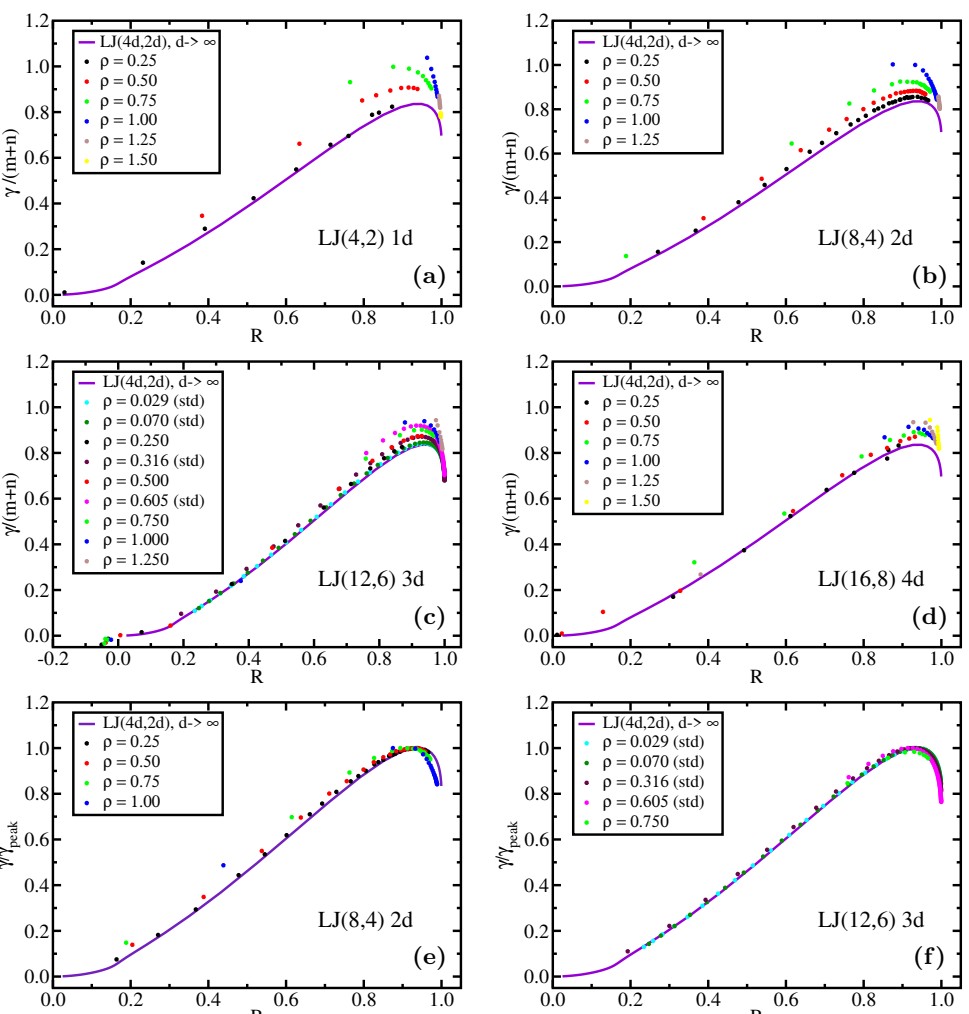

Figure 3: $(R, \gamma)$ diagrams for the LJ$(4d, 2d)$ potential for $d = 1, 2, 3, 4$. The density range is given in the inset while the temperature varies from $T = 0.5 \to 5$ in 0.5 steps (except when otherwise stated). The virial approximation curve Eq. (13) for $n = X = 2$ is represented by the full violet curve. (a) $d = 1$, the potential is $v_{4,2}(r)$. (b) $d = 2$, the potential is $v_{8,4}(r)$. Here the temperature steps are 0.25 starting from $T = 0.25$. For $\rho = 0.25$ and $\rho = 0.50$, the highest temperature simulated is $T = 12.00$ and $T = 7.00$, respectively. This ensures visibility of the peak in $\gamma(R)$ in panels (e) and (f). (c) $d = 3$, the potential is $v_{12,6}(r)$. Data for both the standard 12-6 LJ (labelled "std") and the generalized LJ(12,6) are presented. The unit length for these two potentials is different, resulting in different definitions for the density. The lowest temperature on each isochore is either along the liquid-gas or liquid-solid coexistence curves for the standard LJ; for the generalized LJ(12,6) densities and temperatures are the same as in (a) with the addition of $T = 0.25, 0.75$. The standard LJ phase diagram is therefore explored more comprehensively: the highest temperature on each isochore is roughly 200 times the lowest, well above $T = 5$. Thus we recover the IPL limit $(R, \gamma) \to (1, 4)$. The similarity between panels (a),(b), and (d) demonstrates that covering such a wide region of the phase diagram is not necessary to get an almost complete $\gamma(R)$ curve. (d) $d = 4$, the potential is $v_{16,8}(r)$. (e) and (f) correspond, respectively, to panels (b) and (c) where for each isochore the density-scaling exponent $\gamma$ is scaled by its peak value $\gamma_{\text{peak}}$. The value of $\gamma_{\text{peak}}$ is different on each isochore.

## 4.1 Influence of density and potential exponents on the $\gamma(R)$ relationship

In Fig. 3 we study the $(R, \gamma)$ diagram of the LJ$(4d, 2d)$ ($n = X = 2$) for $d = 1 - 4$. This corresponds to the standard LJ potential for $d = 3$ with rescaled units of length (thus density) and temperature (see Sec. 2). Apart from a rescaling of the density value, the curves $\gamma(R)$ are thus the same for both the standard LJ potential and $v_{12,6}(r)$ in $d = 3$ when exploring the many different state points studied in this work. As discussed in Ref. [25], these curves are similar for all isochores and dimensions. $R$ increases as a function of $T$ in the liquid phase at a fixed density [14], thus temperature increases from left to right along any given isochore. As will become clear from Fig. 4 below, the shape for this particular potential is representative of a wide range of values of $n$ and $X$. At low densities the data are very close to the prediction of Eq. (13) for $n = 2$ and $X = 2$, as expected. As the density is increased, the data stay close to the analytic prediction, although the values of the density-scaling exponent become slightly higher and the state points are more confined to the region around $R = 1$. This is because we do not take into account liquid-vapour coexistence state points, present at low temperature. We are then left with higher temperatures, corresponding to high values of $R$. Note that the attractive term cannot be ignored ($\gamma$ being different from 4). Also, for $d = 1, 2, 3$ ordered phases are found at high densities for which the correlation coefficient $R$ is almost unity. As in the case of phase coexistence, our treatment is not applicable to ordered phases and therefore those state point are not shown.

The quantitative discrepancy at high densities can be reduced through a simple rescaling procedure, using the maximum density-scaling exponent $\gamma_{\text{peak}}$ at each density. This is done in Fig. 3(e)(f) where the gamma exponents for $d = 2$ (panel (b)) and $d = 3$ (panel (c)) have been rescaled by $\gamma_{\text{peak}}$. These two dimensions are displayed as they are the most relevant and because only in these cases, it is possible to clearly identify a maximum value in the $(R, \gamma)$ curve for some isochores. This rescaling accounts almost quantitatively for the density corrections not taken into account by the first-order virial expansion.

The dependence of the previous diagrams on the $n$ and $X$ at fixed dimension $d = 3$ is explored in Fig. 4. In Fig. 4(a) the potential LJ$(2nd, nd)$ is studied by varying $n$ from 4/3 to 6, showing little influence of the value of $n$ on the relation between $\gamma/n$ and $R$, as expected from Eq. (13). In Figs. 4(b)(c)(d) the influence of the ratio $X$ on the LJ$(Xnd, nd)$ potential with $n = 2$ is studied. For moderate values of $X$, little variation is seen; one must reach values of $X \approx 50$ to observe a qualitative shape change. At very high $X \approx 10^3$, the curve converges to the parabolic expression $\gamma = mR^2$ as argued in App. A: the non-parabolic part of the curve gets shifted to the $R = 1$ region as $X$ increases and gradually disappears for $X \gtrsim 50$. There seems to be no distinction in the $(R, \gamma/n)$ diagram between the large-$X$ LJ$(2Xd, 2d)$ and a continuous 'sticky-sphere' potential LJ$(Xnd, nd)$ with $n$ and $X \to \infty$, i.e. , for large $X$ the tail's shape does not matter much in the liquid regime (see also Fig. 1).

## 4.2 Dimensional dependence of the $\gamma(R)$ relationship

We next examine how fast the curves obtained above at finite dimension converge towards the analytical prediction of Eqs. (9) – (16). In order to investigate the deviations from the infinite-dimensional $(R, \gamma)$ diagram, we quantify the deviation by subtracting the $d \to \infty$ curve from the finite-$d$ one, $\gamma_d(R) - \gamma_\infty(R)$, in which $\gamma_d(R)$ is isochore dependent and $\gamma_\infty(R)$ is obtained from Eq. (13) with $n = X = 2$. When increasing $d$, we must compare different isochores in different dimensions, and we shall now detail strategies employed to make this

comparison in a physically meaningful way.

We first plot in Fig. 5 the deviations for $d = 1$ to 4 at fixed density. This choice is natural as the interval of values of the virial-potential energy correlation coefficient plotted in this way is roughly the same for all the dimensions considered. For all densities the deviations decrease as $d$ increases from 1 to 4 (except for the lowest density, Fig. 5(a), where deviations are too low that a trend may be observed and where no big deviations are expected since the virial approximation is exact in the low-density limit). This comparison is somehow not satisfactory because fixing the density means comparing different physical situations. The volume occupied by a particle, $\sim \mathcal{V}_d(\sigma/2)$, where the interaction range of the potential $\sigma$ is interpreted as a particle diameter, decreases monotonically with $d$ and therefore the system gets more diluted as $d$ increases. A plausible interpretation of the reduced deviations with increasing $d$ in Fig. 5 is that the system in $d + 1$ dimensions appears as less dense than its

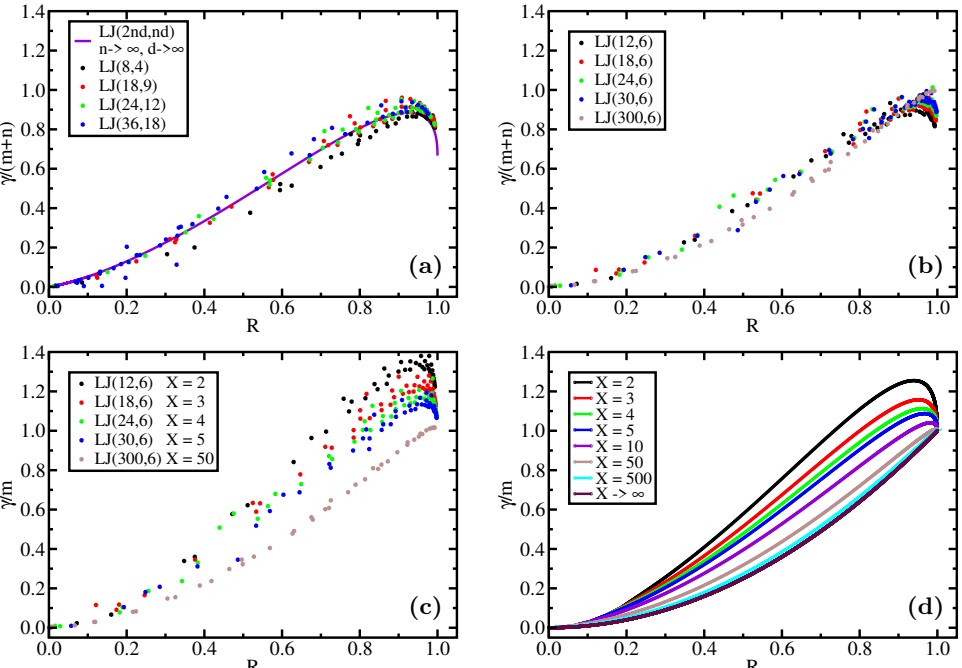

Figure 4: $(R, \gamma)$ diagrams for several LJ potentials in fixed dimension $d = 3$. For each system the state points studied are from the six isochores defined by $\rho = 0.25 - 1.50$ with 0.25 increment, and the temperature range is $T = 0.25 - 1.0$ by steps of 0.25 and $T = 1.0 - 5.0$ by steps of 0.5. For LJ(300,6) only four isochores were simulated ($\rho = 0.25, 0.5, 0.75, 1.0$). State points with negative $R$, corresponding to liquid-vapor coexistence, and state points corresponding to ordered states are not shown for the reasons discussed in Sec. 4.1. (a) LJ($2nd, nd$) with $n \in \{4/3, 3, 4, 6\}$. The solid line is the $n \to \infty$ virial prediction Eq. (15). These data were first presented in Ref. [25]. (b) (c) (d) LJ($Xnd, nd$) ($d = 3$, $n = 2$) for several ratios $X$. (b) For $X = 2, 3, 4, 5, 50$, $R$ versus the rescaled $\gamma/(m + n)$ vary little. For $X > 50$ the shape changes qualitatively. The infinite-temperature $R = 1$ endpoint varies slightly with $X$, as $\gamma/(m + n) = X/(X + 1)$, panels (c),(d). In App. A we show that when $X$ becomes large, the natural scaling variable is $\gamma/m$. This is seen both in MD simulations (c) and from Eqs. (13) (d).

counterpart in dimension $d$. As a result it could be that the virial truncation performs better not because the mean-field $d \to \infty$ approximation improves, but primarily as an indirect 'low-density' effect. To compensate for this, the density must increase when comparing the $d$ to $d+1$ data, as has been observed for the liquid-crystal transition [28, 46] or the liquid-glass transition [47, 49]. Note that in the large-dimensional limit, the dense liquid region emerges for densities scaling exponentially in $d$ [36, 37, 39, 40, 44].

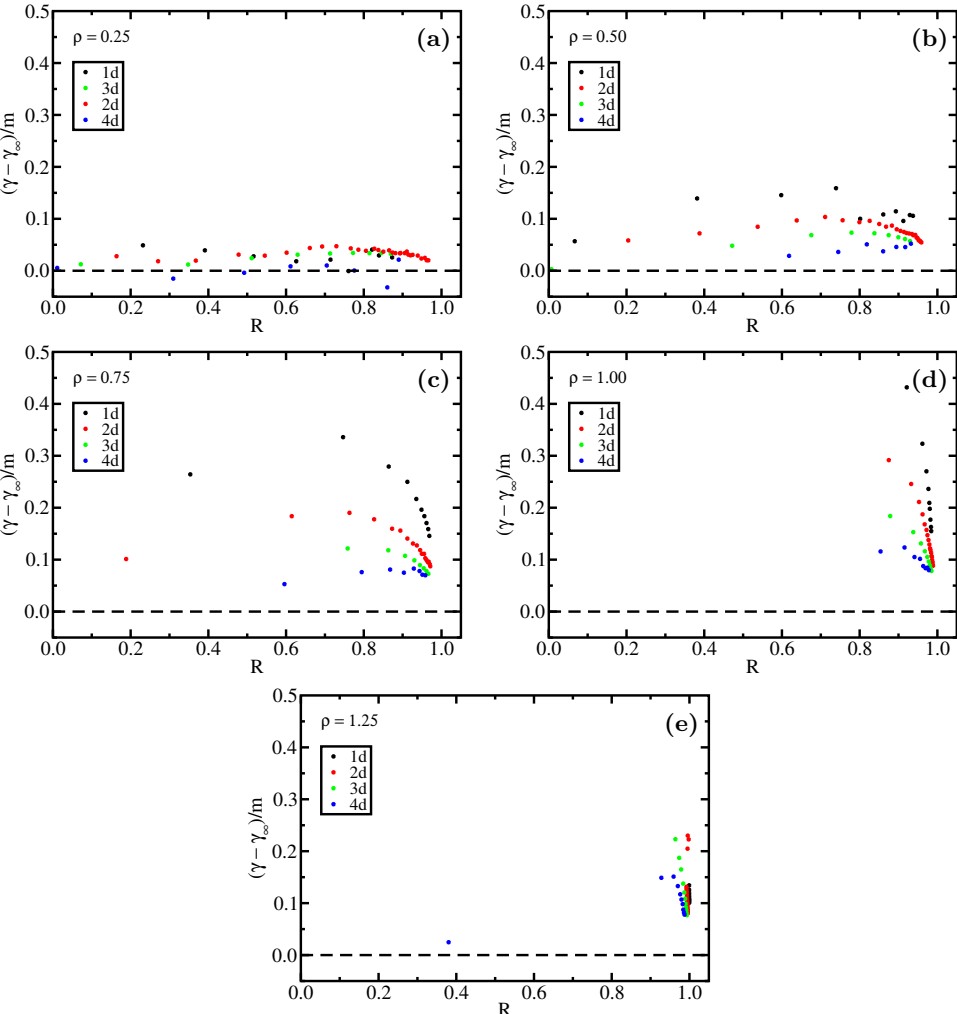

Figure 5: Relative deviations from the $d \to \infty$ value of the $(R, \gamma)$ curves of the LJ$(4d, 2d)$ systems for dimensions $d = 1$ to 4. In each plot we have subtracted the $d \to \infty$ curve obtained in Eq. (13), $\gamma_\infty(R)$, from the simulation data $\gamma_d(R)$, rescaled by $m \; (= 4)$. The horizontal dashed line at the origin of the vertical axis corresponds to the exact $d \to \infty$ curve. Each plot from (a) to (e) is given for a fixed isochore with density varying from $\rho = 0.25$ to 1.25 in steps of 0.25. The higher the density, the closer are state points from the $R = 1$ boundary. Simulation data before subtraction are displayed in Fig. 3. The value of $\gamma$ estimated from Eq. (13) is a lower bound to the actual value found from simulations, at least away from the low-density limit where Eq.(13) becomes exact.

There is no *a priori* simple way to compute densities of related physical regimes in different dimensions. In order to get reasonable values, we attempted two different strategies, focusing on fairly dense liquid regimes, since their large-dimensional limit is well understood [27, 37, 40, 44]. The first strategy computes ratios between the dynamical glass transition densities $\rho_{\mathrm{d}}^{\mathrm{HS}}(d)$ for hard spheres. This is the only potential for which such a transition has been determined in dimensions ranging from 2 to 12 [49, 53]; temperature does not influence this scaling. The second strategy amounts to estimating as a function of the dimension the density at which the first-order virial expansion breaks down, providing meaningful density ratios for the regime we are after. This characteristic density, denoted by $\rho_{\mathrm{Z}}(d,T)$, is calculated by comparing the first- and second-order virial coefficients, yielding $\rho_{\mathrm{Z}}(d,T) \sim |B_2(T)/B_3(T)|$ in which $B_{2,3}(T)$ are the second and third virial coefficients, respectively [24]. The density ratios between different dimensions are fairly independent of temperature far from liquid-gas coexistence (App. C). The dimension-dependent density ratios provided by both methods are listed in Tab. 2.

|  | $\rho_{d=2}/\rho_{d=1}$ | $\rho_{d=3}/\rho_{d=2}$ | $\rho_{d=4}/\rho_{d=3}$ |
|---|---|---|---|
| HS dynamical transition | 1.019 | 1.071 | 1.192 |
| Virial corrections | 1.13 | 1.23 | 1.30 |

Table 2: Density ratios used in Fig. 6 to compare different dimensions from $d = 1$ to 4. The first method (first line) gives ratios between the dynamical transition densities for hard spheres $\rho_{\mathrm{d}}^{\mathrm{HS}}(d)$ extracted from Refs. [49, 53]. One-dimensional hard spheres do not exhibit such a slowing down, so for $d = 1$ we considered the dense regime to occur at the maximal packing fraction (unity). The second method (second line) is based on the calculation of $\rho_{\mathrm{Z}}(d,T)$ (App. C).

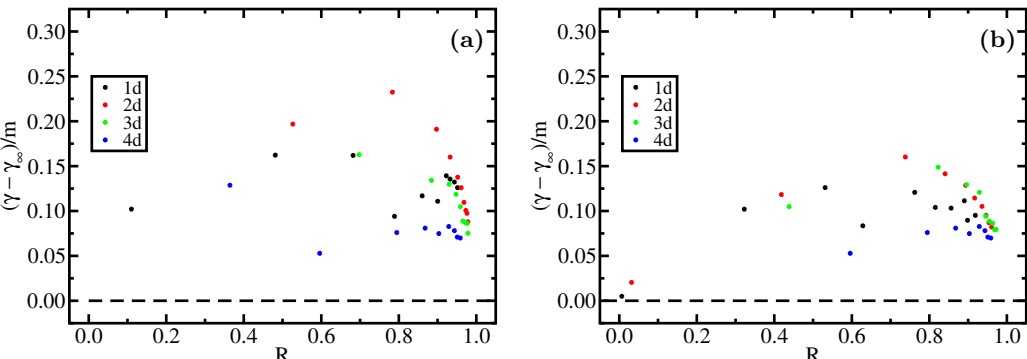

Figure 6: Deviations from the $d \to \infty$ value of the $(R,\gamma)$ curve of the LJ($4d, 2d$) potential for dimensions $d = 1$ to 4, using dimension-scaled densities. (a) Densities scaled by the hard-sphere dynamical transition density $\rho_{\mathrm{d}}^{\mathrm{HS}}(d)$ (first line of Tab. 2); $\rho_{d=1} = 0.577$, $\rho_{d=2} = 0.587$, $\rho_{d=3} = 0.629$, and $\rho_{d=4} = 0.750$. (b) Densities scaled by $\rho_{\mathrm{Z}}(d,T)$ obtained by equating the second and third virial expansion terms (second line of Tab. 2). Specifically, $\rho_{d=1} = 0.415$, $\rho_{d=2} = 0.469$, $\rho_{d=3} = 0.577$ and $\rho_{d=4} = 0.750$.

Both scaling-method outcomes are plotted in Fig. 6. We fixed the density in $4d$ to be $\rho_{d=4} = 0.75$, an intermediate value. This value of density is not so low that meaningful measurable differences with respect to the $d \to \infty$ analytic $(R, \gamma)$ curve are detected, and not so high to span a large interval of the computed virial-potential energy coefficient (Fig. 5). In the hard-sphere dynamical transition scaling, we observe a convergence to the $d \to \infty$ analytic prediction going from $d = 2$ to $d = 4$. The one-dimensional values are, however, somewhat closer to the large-$d$ result than the two- and three-dimensional ones. The virial scaling, which gives much lower densities for each dimension $d \leqslant 3$, still yields qualitatively similar results, albeit with almost no difference between the $d = 2$ and 3 cases. We conclude that, similarly to the non-scaled plots of Fig. 5, already for low dimensionality, a convergence towards the large-dimensional $(R, \gamma)$ diagram is observed.

## 5   Conclusion

We have introduced a set of simplified equations to compute the virial-potential energy correlation coefficient $R$ and the density-scaling exponent $\gamma$, valid in any dimension and for any pair potential in the isotropic liquid phase. These equations are obtained from a low-density virial expansion. As such they are exact in two limits: the low-density limit (in a given dimension) and the infinite-dimensional limit (for any density), in the isotropic liquid.

We have specialized these results to the case of $\mathrm{LJ}(m', n')$ systems. The interest of such equations is that both $R$ and $\gamma$ can be computed straightforwardly, through numerical integrations of a few one-dimensional integrals, for any state point in the phase diagram. We showed through molecular dynamics simulations that this approximation applies qualitatively and almost quantitatively for the $(R, \gamma)$ diagram, if the system does not phase separate – a case in which density scaling does not apply, even approximately. It appears that density corrections are weak for the $\mathrm{LJ}(m', n')$ potentials, irrespective of the value of the exponents or temperature. The analytical shape $\gamma(R)$ allows one then to make the following prediction, robust in the whole fluid phase: if the measured exponent $\gamma$ decreases when increasing temperature along an isochore, then we are probing a strongly-correlating regime, i.e. a regime where density scaling is satisfied to a good approximation [25].

We do not expect the monotonicity of $\gamma$ to yield a direct indication of good scaling for any potential. Yet, the simplified low-density limit expressions from Eq. (9) can be helpful for other potentials, as they provide the relation $\gamma(R)$, allowing one to assess $R$ from the possible measurement of $\gamma$ [17,23], or the $R$–$\gamma$ relation if the latter function is multivalued (this occurs for instance for a potential consisting in a sum of IPL potentials with different exponents). As an example, preliminary data shows that the function $\gamma(R)$ for WCA potentials [24,54] gives the correct qualitative behaviour while significantly differing in the functional form from the $\mathrm{LJ}(m', n')$ one.

It was recognized in Ref. [27] that perfect density scaling is achieved for many non-trivial potentials – i.e., potentials which are not necessarily Euler homogeneous – in the $d \to \infty$ limit. $\mathrm{LJ}(m', n')$ potentials do not display such a perfect scaling in the high-$d$ limit, and are instead characterized by a $(R, \gamma)$ diagram not restricted to a single point. We varied the dimension from $d = 1$ to $d = 4$ in order to check the convergence to the high-$d$ $(R, \gamma)$ diagram for intermediate densities where there are corrections to the low-density expressions. Using several possible scalings of densities with dimension leads to the same conclusion of a

monotonic shrinking of fluctuations from $d = 2$ to 4. We interpret this as yet another instance of the large-dimensional being qualitatively and sometimes even quantitatively good [37] for low dimensions.

## Acknowledgements

T.M. warmly thanks the *Glass and Time* laboratory of Roskilde University for their hospitality. The authors thank Francesco Zamponi for discussion.
Data is available online on the *Glass and Time* repository www.glass.ruc.dk/data/.

**Funding information** This work was supported by the VILLUM Foundation *Matter* grant (16515) [J.C.D] and by a research grant (00023189) from VILLUM FONDEN [L.C].

## A  Large-exponent ratio limit

Here we investigate the large-$X$ limit ($X = m/n$) of Eq. (13). As argued in Sec. 3.1, the $n$ dependence is rather mild, so that we send first $n \to \infty$. From Eq. (15) we thus need to study the large $X$ limit of $I_U(X, \beta\epsilon)$ and $I_W(X, \beta\epsilon)$. Consider the latter: From the behaviour of the integrand at large $X$ one finds that only $y^X \leqslant 1$ contributes and the integral can be roughly approximated by

$$I_U(X, \beta\epsilon) \underset{X \to \infty}{\sim} X^2 \int_0^1 \mathrm{d}y\, y e^{\beta\epsilon y} = \left(\frac{X}{\beta\epsilon}\right)^2 \left[1 + (\beta\epsilon - 1)e^{\beta\epsilon}\right] \tag{18}$$

$I_W(X, \beta\epsilon)$ is calculated from the contribution of its saddle point $y^{\mathrm{sp}} > 1$ defined by the competition between $y^{2X}$ and $e^{-\beta\epsilon y^X / X}$, i.e. $y^{\mathrm{sp}} = \left(\frac{2X}{\beta\epsilon}\right)^{\frac{1}{X}}$. From the Laplace method [55] at large $X$ one gets, up to a numerical prefactor,

$$I_W(X, \beta\epsilon) \propto \frac{X}{(\beta\epsilon)^2} e^{\beta\epsilon} \, , \tag{19}$$

a scaling that is well verified numerically for all temperatures. From (15) we arrive at

$$R \propto \frac{1}{\sqrt{X}} \frac{e^{-\beta\epsilon/2}}{\sqrt{1 + (\beta\epsilon - 1)e^{\beta\epsilon}}} \qquad \text{and} \qquad \frac{\gamma}{m} = \frac{1}{X} \frac{1}{1 + (\beta\epsilon - 1)e^{\beta\epsilon}} \tag{20}$$

which yields $\frac{\gamma}{m} \propto e^{\beta\epsilon} R^2$. This holds for $\beta\epsilon$ not scaled with $X$, smaller than any power of $X$, while a scaling such that $\beta\epsilon \ll 1$ is still described by this saddle point. Therefore, since in this limit $\beta\epsilon \ll 1$ (i.e. $T \to \infty$) one has $\gamma = m$ and $R = 1$ as the repulsive IPL dominates (see Eq. (10)), we expect that the relation

$$\frac{\gamma}{m} = R^2 \tag{21}$$

approximates well the curve and should be a lower bound to all finite $X$ curves. Since both $\gamma$ and $R$ go to zero for $X \to \infty$, the non-trivial part of the finite-$X$ curves gets shifted to higher temperatures, i.e., closer to $R = 1$, and washed out in the large-$X$ limit.

## B  Estimating $R$ and $\gamma$ in computer simulations: influence of simulation length and system size

In this Appendix, the statistical errors on $R$ and $\gamma$ are reported for a fixed liquid-state point $(\rho, T) = (1.0, 2.0)$ with potential $v_{16,8}(r)$ in $d = 4$ for several choices of simulation length $N_{\mathrm{steps}}$ and system size $N$. We here investigate in detail $d = 4$ simulations as they are the most time consuming: we aim at a fast enough simulation while maintaining a large enough $N$ (and $N_{\mathrm{steps}}$) for accurate computation of observables. We show that even short simulations (a few million time steps) and relatively small system sizes can lead to a good estimation of $R$ and $\gamma$, a well-known fact for practitioners although rarely explicit. The statistical errors on every state point studied in this work are available online (see Acknowledgements).

Fig. 7 displays the average value of $R$ (a) and $\gamma$ (b) as a function of the simulation length for a fixed system size $N = 4096$. Each simulation is divided into 5 blocks and the error bars

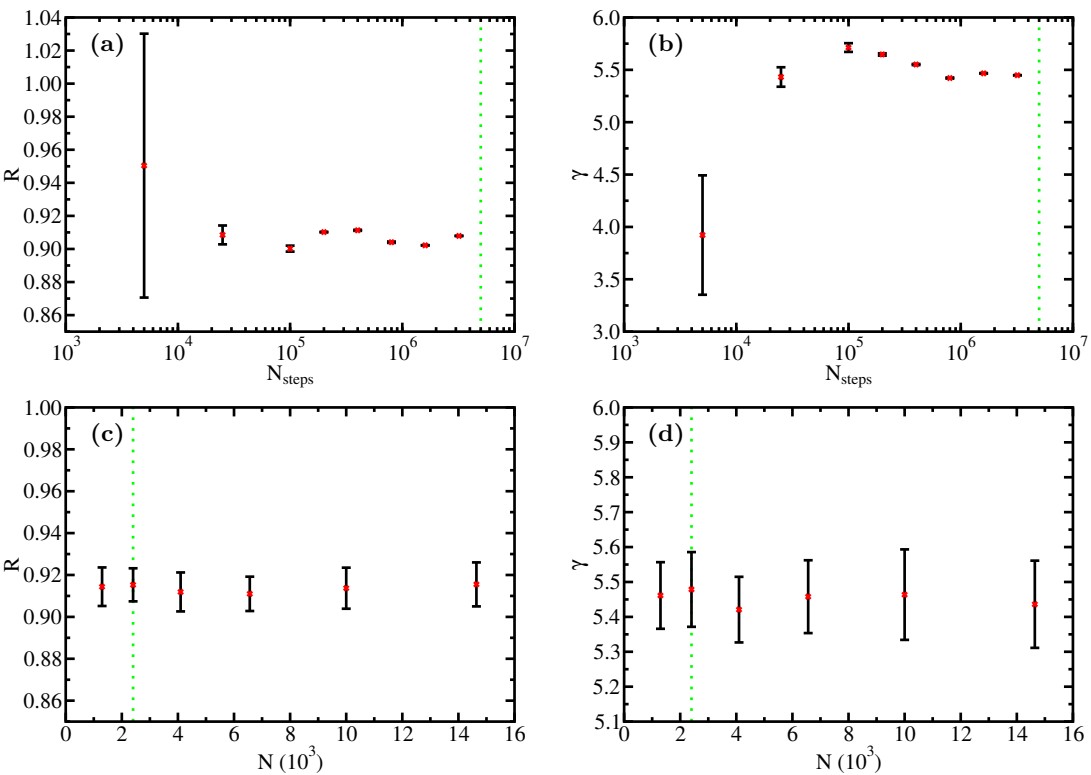

Figure 7: Influence of simulation length $N_{\text{steps}}$ (a)(b) and system size $N$ (c)(d) on the estimated values of $R$ and $\gamma$. The green vertical dotted line indicates the minimum simulation length (a)(b) or smallest system size (c)(d) of the article's $d = 4$ data.

are obtained as standard deviation of the mean values obtained from each block. $d = 4$ data in this article used between 50 and 120 blocks depending on the simulation length. The reduced time step $\Delta \tilde{t} = 0.004$ is fixed (for all four panels). Both energy and virial instantaneous samplings (required for computing $R$ and $\gamma$) are saved every 200 time steps.

The figure exhibits as well the average value of $R$ (c) and $\gamma$ (d) as a function of the system size, for the same state point and potential. The simulation length is fixed as $N_{\text{steps}} = 5 \cdot 10^6$ and each simulation is divided into 50 blocks. The smallest system size analyzed here ($N = 1296$) corresponds to a box of linear size $L = 6.0\sigma$. Note that all potentials studied in this work are cutoff at $r_{cut} = 2.5\sigma$, therefore system sizes corresponding to $L < 5.0\sigma$ cannot be considered.

## C Dimensional scaling at dense liquid densities derived from the virial expansion

In this Appendix we elaborate on the scaling of density used in Fig. 6(b) for the $\mathrm{LJ}(4d, 2d)$ potential. The virial expansion of the (liquid) equation of state is [24]:

$$\beta P = \rho + B_2(T)\rho^2 + B_3(T)\rho^3 + O(\rho^4)$$

$$B_2(T) = -\frac{1}{2}\int \mathrm{d}\mathbf{r}\, f(r) \quad \text{with} \quad f(r) = e^{-\beta v_{4d,2d}(r)} - 1 \tag{22}$$

$$B_3(T) = -\frac{1}{3}\int \mathrm{d}\mathbf{r}\mathrm{d}\mathbf{r}'\, f(r)f(r')f(|\mathbf{r} - \mathbf{r}'|)$$

In this paper we have considered only first-order virial expansions; these apply at low density and/or high dimension. In finite dimensions one expects the first-order approximation to break down for densities at which the next term is relevant, i.e. whenever $B_2(T)\rho^2 \approx B_3(T)\rho^3$. This defines a density $\rho_{\mathrm{Z}}(d,T) = -B_2(T)/B_3(T)$. At this density, the compressibility factor becomes $\beta P/\rho_{\mathrm{Z}} = 1 + O(\rho_{\mathrm{Z}}^3)$, meaning that $\rho_{\mathrm{Z}}(T)$ coincides with the low-density Zeno[6] line [56, 57], which in $d = 3$ lies inside the supercritical region well above the liquid-vapor critical point.

   We computed numerically $\rho_{\mathrm{Z}}(T)$ for $d = 1$ to 4 (Fig. 8(a)). Both virial coefficients are negative at low $T$, and become positive above it. $B_2(T)$ vanishes for a dimension-independent value, the Boyle temperature $T_{\mathrm{Boyle}} \simeq 3.4$ (as can be readily seen from the definition in Eq. (22) from the same manipulations as the ones in Sec. 3.1) [57, 58], whereas $B_3(T)$ vanishes close to $T = 1$ depending on the dimension. This explains the observed divergence of $\rho_{\mathrm{Z}}(d, T)$ in this region. The negativity of these coefficients indicates phase separation at small enough density, i.e. here a liquid-vapor coexistence. Indeed from Eq. (22) at small density this negativity implies $\mathrm{d}P/\mathrm{d}\rho < 0$, which signals a thermodynamic instability. As mentioned in Sec. 4 we wish to stay away from this regime in which our analytical results are no longer justified. The low-density equation of state Eq. (22) shows no sign of phase separation above $T \simeq 1.7$ for any dimension $d = 1$ to 4. Thus focusing on these higher temperatures, corresponding to the fluid phase above coexistence (at small enough densities), one realizes that all the density ratios at fixed temperature $\rho_{\mathrm{Z}}(d + 1, T)/\rho_{\mathrm{Z}}(d, T)$ are approximately constant, compare Fig. 8(b). Note that as we are interested in an order of magnitude for density ratios between different dimensional systems for which the first virial truncation breaks down, the sign of $\rho_{\mathrm{Z}}(d, T)$ does not matter. Consequently, we can extract a temperature-independent meaningful scaling of density by averaging the value of this ratio over the whole temperature range $T \in [T_0, 5]$. We took $T_0 = 1.96 > 1.7$ as the choice of $T_0$, which does not modify considerably the ratio values while we must at the same time consider enough statistics, as displayed in Figs. 8(c)-(d). For this value of $T_0$, indeed, fluctuations of the computed density ratio are below 5% with respect to the average in all dimensions. This procedure provides the numbers indicated in the last line of Tab. 2, which are appreciably above the density ratios defined by the hard-sphere dynamical transition densities (first line). The choice of $T_0$ close to 1.7 maximizes the ratio values with respect to higher temperature (see Figs. 8(b)-(c)). As higher densities are associated with stronger deviation from the virial approximation (compare Fig. 3), this is the most unfavorable situation in order to see smaller deviations to the large-$d$ curve $\gamma_\infty(R)$; yet we find in Fig. 6 a good convergence for $d = 2 \to 4$ using such ratio values.

---

[6]The Zeno line is the line in the $(\rho, T)$ phase diagram of state points at which the virial is zero.

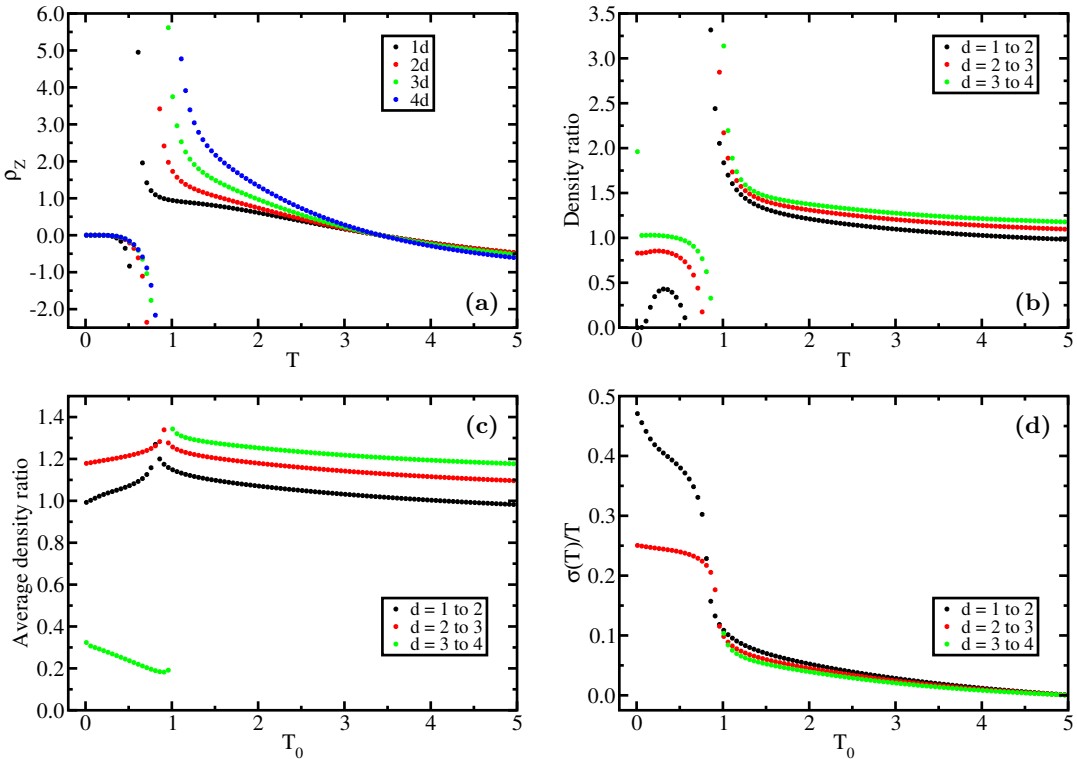

Figure 8: Density scaling between similar liquid regimes in different dimensions $d = 1$ to 4. For low enough densities there is a liquid-gas phase coexistence below $T \simeq 1.7$ for $d = 4$ (for lower dimensions this temperature gets slightly lowered). (a) Evolution of the Zeno-like density $\rho_Z = -B_2/B_3$ with temperature. (b) Density ratios between consecutive dimensions $\rho_Z(d+1, T)/\rho_Z(d, T)$. (c) Dependence on the lower boundary temperature $T_0$ of the averaged density ratios between consecutive dimensions $\langle \rho_Z(d+1, T)/\rho_Z(d, T) \rangle_T$. The average is over temperature on the range $T \in [T_0, 5]$. (d) Measure of the fluctuations in the temperature average of Fig. (c) (standard deviation $\sigma_T$ over the average value $\langle \rangle_T$ of the density ratios in (b)) as a function of the lower boundary temperature $T_0$. For $T_0 = 1.96$ the fluctuations are less than 5% of the average for all dimensions considered. Data below $T_0 < 1$ for $d = 3 \to 4$ is not shown as it fluctuates more widely (standard/mean $= 5 - 6$).

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
