# Peer review of "Density scaling of generalized Lennard-Jones fluids in different dimensions"

_SciPost Physics_

## Round 2 · Referee Report · Anonymous · 2020-6-29

Strengths

1) This work further formalizes the theoretical context for the density scaling approach popularized by one of the co-authors.
2) The theory is assessed by a numerical study that takes the dimensional dependence into account.

Weaknesses

1) The organization of some of the figures does not always help with legibility.

Report

After the requested changes have been made, this manuscript should meet the publication criteria for this journal.

Requested changes

1) Presumably, the choice of normalization of Eq. (3) (to ensure "that the minimum does not shift when exponents vary") is made for some analytical or definitional convenience. Please specify.

2) Novel, dedicated figures should be included to enhance the discussion of Eqs (13) and (16). For instance, the mild dependence on n is not obvious to tease out from the equations alone. The non-trivial shape of the parametric temperature plot would also be more easily appreciated if it were presented separately. Panel (d) from Fig. 3 could additionally be included as part of this new presentation.

3) Figures should be discussed in the order in which they appear. A proposal: for ease of comparison, Figs 2 and 4 should appear side by side and the panels Fig. 6 might be included with those of Fig. 2.

4) The d=1 results could presumably also be approximated by a transfer matrix approach with next-nearest neighbour interactions, see, e.g., M. Godfrey and M. Moore, Phys. Rev. E 91, 022120 (2015). A note to this effect should be included.

5) On p. 11, the simulations are stated to be sufficiently long and sufficiently large for thermodynamic quantities to be accurately calculated. Are these estimates based on published data on quantitative convergence tests, or on length scale analysis? Additional details should be provided to motivate the "running time limitations".

Small correction:
i) On p. 4, kB is set to unity, and the quantity is redefined on p. 6.

---

## Round 2 · Referee Report · Anonymous · 2020-7-10

Strengths

- Solid theoretical and numerical work
- The derived analytical expressions may provide useful guidelines to simulations and experiments on dense liquids

Weaknesses

- Some parts of the presentation can be improved

Report

The paper presents analytical expressions for the density scaling exponent gamma and pressure-energy correlation coefficient R for generalized Lennard-Jones models. The expressions, which are exact in the low density or high-dimensional limit, are tested against molecular dynamics simulations in spatial dimensions from 1 to 4. The agreement is qualitative, but it improves as d increases, provided that potential parameters and state variables are scaled appropriately. The analytical results provide a guideline to understand the trends of gamma and R across the phase diagram and by changing spatial dimension. Overall this is valuable contribution, which I recommend for publication in SciPost provided the minor points below are taken into account.

Requested changes

- I am not sure if this was discussed in some previous paper of the authors, but what is the physical interpretation of the peak in gamma(R)?

- p.11: "In 4d the system size is chosen to be the smallest necessary
to reliably study the highest-density state points (ρ = 1.5)."

What kind of tests did the author perform to ensure this? Please mention it in the text.

- p.13 "Also, for d = 1, 2, 3 ordered phases are found at high densities for which
the correlation coefficient R is almost unity."

Which points in the graph correspond to ordered states? It would be helpful to mark them somehow.

- fig.2 and 3 : the captions are pretty long; also part of the discussion in the caption would fit better in the main text

- fig.6a : y label should be gamma/gamma_peak

- fig.6: I think fig.6 should appear right after fig.2. Indeed, the discussion of fig.6 follows immediately the one of fig.2

---

## Editorial Decision

editor-in-charge_assigned